# Clusterin Regulates the Mechanisms of Neuroinflammation and Neuronal Circuit Impairment in Alzheimer’s Disease

**DOI:** 10.3390/ijms26157271

**Published:** 2025-07-28

**Authors:** Yihang Yu, Chunjian Wang, Binbin Wang, Xuelin Wang, Qain Zhao, Yan Yan, Xiaoyun Liu

**Affiliations:** 1School of Basic Medical Sciences, Hebei Medical University, Shijiazhuang 050017, China; 18000652260@163.com (Y.Y.); wangcj_wang@163.com (C.W.); 2The First Hospital of Hebei Medical University, Shijiazhuang 050031, China; 28402060@hebmu.edu.cn (B.W.); wangxuelin0405@163.com (X.W.); drzhao163_2025@163.com (Q.Z.)

**Keywords:** Alzheimer’s disease, clusterin, neuronal circuit

## Abstract

Alzheimer’s disease (AD) is a neurodegenerative disease with a multifaceted pathogenesis, which remains elusive, seriously affecting the quality of life of elderly patients and placing a heavy burden on affected individuals, their families, and society. As third-party synapses in brain networks, astrocytes play an important role in maintaining the normal function of neural networks, which contribute to the abnormal function of networks in AD. In recent years, numerous studies have shown that clusterin, a protein expressed by astrocytes, can participate in the progression of AD. Clusterin plays a significant role in many pathological processes of AD, such as lipid metabolism, AD pathological features, the imbalance in neural circuit excitatory inhibition, and neuroinflammation. Therefore, delving deeper into the association between clusterin and AD will help us to understand the mechanisms of disease better and provide a theoretical basis for early diagnosis and the development of treatment strategies for AD.

## 1. Introduction

Astrocytes constitute approximately 30% of the cells in mammals’ central nervous system (CNS), which are integral components of the brain, playing a crucial role in various physiological functions that are essential for normal neural development, synaptic formation, and the proper propagation of action potentials. Astrocytes maintain synaptic homeostasis through various physiological functions, including calcium homeostasis (uptake, recycling, and release of neurotransmitters), synaptic activity and ion balance regulation, and energy and lipid metabolism [1]. Reactive astrocytes observed in various neurodegenerative diseases refer to the morphological, molecular, and functional reshaping of astrocytes in response to challenges in the brain, such as the deposition of amyloid-beta [Aβ] or α-synuclein [α-syn], infection, damage, or the degeneration of neurons [2,3]. Protein dysfunctions in astrocytes in neurodegenerative diseases may constitute a pivotal factor underlying these conditions and could potentially emerge as critical therapeutic targets for various neurodegenerative diseases, including AD. Several astrocyte-specific risk factors have been reported, such as in regard to the well-known glial fibrillary acidic protein (GFAP). In clinical studies of AD, the GFAP in cerebrospinal fluid and blood is extensively measured and considered to be a biomarker for reactive astrocyte proliferation [4,5,6]. Extensive clinical studies suggest that the increased expression of blood GFAP in AD patients can serve as a target for early diagnosis of AD. GFAP levels were significantly increased in AD patients compared to patients without dementia, suggesting that blood GFAP concentrations rose at a higher rate in people who would go on to develop AD [4,7,8] (Table 1). However, the peripheral source of GFAP also affects blood GFAP levels, which may affect the accuracy of blood GFAP concentrations for the early diagnosis of AD [9,10,11], considering that the concentration of GFAP in the blood is also related to the degree of amyloid-β (Aβ) degradation and the phosphorylation of tau. Diagnosis based on a combination of the concentration of GFAP in the blood with the degree of Aβ degradation and the phosphorylation of tau is used as a composite indicator for patients with early MCI [12]. In addition, GFAP can be used specifically to predict the onset of dementia, and repeated measurements of GFAP in the preclinical phase of dementia may be more valuable as an indicator of future treatment efficacy in a health clinic. Clinical studies show that the increased expression of blood GFAP can be used as an optimal biomarker for dementia prediction, more than 10 years before diagnosis [9]. And the elevation in plasma GFAP levels can be used to identify cognitively normal elderly individuals at risk of AD [13].

Undoubtedly, the pivotal role of astrocytes in AD is increasingly recognized [14,15,16]. The application of soluble Aβ oligomers (Aβo) induced fast and widespread calcium hyperactivity in the astrocytic population and in the microdomains of the astrocyte arbor. Interestingly, in an AD mouse model (APP/PS1 mouse), astrocyte calcium hyperactivity equally takes place at the beginning of Aβ production, depends on TRPA1 channels, and is linked to CA1 neuron hyperactivity [17]. Similarly, the disruption of neural networks in early AD is accompanied by a decrease in astrocyte calcium signaling. The restoration of calcium activity in astrocytes normalizes the functional connectivity of neuronal networks [18]. Currently, understanding the regulatory mechanisms of astrocytic risk factors in AD, such as GFAP [13,19,20], tau [20,21], and ApoE [22,23,24,25], stands as a prominent area of research focus. However, the clusterin protein (later referred to as Clu), regulated by the *Clu* gene, is a protein-coding gene involved in various cellular processes, including cell survival, lipid metabolism, and the immune system, which may also be a new risk factor for AD [15,26,27,28]. The Rs11136000, Rs93318888, and Rs2279590 variants in the *Clu* gene represent the genetic risk factors for late-onset AD [27,29,30]. The Rs11136000 variant is considered to be the most important AD risk factor, located in the third intron of the *Clu* gene, and the Rs11136000-C allele can increase the probability of AD by 1.16 times [27,30]. Both clinical studies on AD cohorts and basic research using an AD mouse model reveal changes in the expression levels of *Clu* in the diseased state of AD, indicating a robust correlation with AD’s pathological features [15,26,27,28]. However, the regulatory mechanisms of *Clu* during the development of AD remain elusive. Exploring the mechanistic role of *Clu* in AD, characterized by its astrocytic-specific expression, stands poised to deepen our comprehension of AD’s pathogenic pathways and may yield insights into discerning and targeting precise therapeutic targets.

In summary, recent technological progress has unveiled the substantial involvement of astrocytes in neurodegenerative disorders. Within this review, we delineate the research trajectory concerning astrocytes in AD and concentrate on elucidating potential mechanisms underlying the astrocyte-specific risk factor, *Clu*, in AD. These mechanisms encompass lipid metabolism, AD pathological hallmarks, disruptions in neural circuit excitatory inhibition, and the onset of neuroinflammation.

## 2. Research History on the Role of Astrocytes in AD

Astrocyte research spans nearly 200 years, evolving from the proposal on neuroglia to the establishment of the concept of astrocytes, exploring their morphology to elucidating their fundamental functions. Subsequently, with technological advancements, the understanding of astrocyte functions has gradually improved, encompassing optogenetics, electrophysiology, single-cell sequencing, microscopic imaging, and others (Figure 1).

### 2.1. Astrocyte Morphology and Calcium Signaling

As far back as 1846, Rudolf Virchow, revered as the father of modern pathology, introduced the concept of “neuroglia”, articulating his thoughts on this subject in a lecture delivered in 1858. Since then, the term neuroglia and its underlying concepts has spread worldwide [31]. In 1865, Deiters first discovered non-neuronal cells in central nervous tissue [32]. In 1893, Michael von Lenhossek introduced the term “astrocyte” and, soon after, Kölliker and Andriezen categorized them into fibrous and protoplasmic astrocytes [33]. Rudolf Virchow first described neuroglial cells in the mid-19th century. It was not until 1888 that a seminal review by Somjen GG synthesized the findings of Santiago Ramon y Cajal, Camilo Golgi, and Pio del Rio Hortega, who, using microscopy and accurate histological staining, described the complex morphology of astrocytes and revealed their diversity [32]. Astrocytes play pivotal roles in the brain, encompassing the promotion of the blood–brain barrier, synaptogenesis, ion homeostasis, neurotransmitter buffering, and the secretion of neuroactive substances, contributing to fundamental physiological functions within the central nervous system [34]. In 1898, Carl Ludwig Schleich proposed the concept of interactions between astrocytes and neurons, suggesting that astrocytes could move between nerve cells, potentially inhibiting neuronal signal transmission [35]. Furthermore, in 1990, there was a groundbreaking discovery in calcium (Ca^2+^) signaling imaging of astrocytes. Astrocytes were found to respond to neuronal synaptic activity by increasing intracellular Ca^2+^ signals through an increase Ca^2+^ concentrations [36].

### 2.2. Astrocyte Typing

Since the advent of single-cell technology in 2009, a growing array of astrocyte cell atlases has emerged, providing high-resolution spatial maps of gene activity for all types of cells [37,38]. To understand the diversity, regulation, and function of astrocyte reactivity, several studies have employed quantitative genomic analysis to characterize various astrocyte gene expression profiles [39,40]. Through the use of purification and genetic analysis of reactive astrocytes, it was found that a systemic lipopolysaccharide (LPS) injection (generating a neuroinflammatory model) and middle cerebral artery occlusion (MCAO, a focal cerebral ischemia model) induced two different types of reactive astrocytes, now known as A1 and A2 astrocytes (similar to the M1/M2 macrophage and microglia paradigm) [39,41]. Although the simple dichotomy of the A1/A2 phenotype does not reflect the broad range of astrocyte phenotypes, it is useful to understand the response status of astrocytes to disease. The terms A1 and A2 astrocytes were widely used in astrocyte research, categorizing these two types into opposing roles: A1 acts in a detrimental way, while A2 plays a beneficial role. However, the latest developments in single-cell technology have revealed various astrocyte states associated with specific developments, aging, and disease processes in human and mouse brains. For example, single-cell RNA-seq (scRNA-seq) or single-nucleated RNA-seq (snRNA-seq) enables the identification of astrocyte clusters in AD states, by analyzing their transcriptional characteristics. The results of single-cell sequencing show that specific proteins for A1 astrocytes include Lcn2, Steap4, S1pr3, Timp1, Hspd1, Cxcl1o, Cd44, Osmr, Cp, Serpina3n, Aspg, Vim, GFAP, H2-T23, Serping1, ligp1, Gbp2, Fbln5, Ugt1a1, Fkbp5, psmb8, Srgn, and Amigo2, and, for A2 astrocytes, they include Clcf1, Tgm1, Ptx3, S100a10, Sphk1, Cd109, ptgs2, Emp1, Slc10a6, Tm4sf1, B3gnt5, and cd14 [41]. However, this simplistic classification fails to capture the complexity of astrocytes, particularly in the context of neurodegenerative diseases. The single-cell sequencing results for an AD mouse model show that low GFAP expression in AD is mainly associated with Gpc5, Luzp2, Lsamp, Trpm3, Slc7a10, and others [15], while the high GFAP state in AD is primarily associated with Apoe, Clu, Ckb, Cpe, Gpm6b, CD44, TNC, HSPB1, HSP90AA1, SLC1A2, SLC1A3, GLUL, SLC6A11, NRXN1, CADM2, PTN, GPC5, and others [15,26]. The high expression of CD44 and TNC, which are both involved in interactions with the extracellular matrix, along with the elevated expression of HSPB1 and HSP90AA1, which are both partners in protein homeostasis, are notable markers [26]. Among them, genes related to glutamate/GABA homeostasis, such as SLC1A2, SLC1A3, GLUL, and SLC6A11, and genes associated with synaptic adhesion/maintenance, such as NRXN1, CADM2, PTN, and GPC5, indicate the loss of homeostatic functions [26]. The transition from low to high GFAP expression in AD is primarily associated with GPC5, Lucp2, Lsamp, Trpm3, Nnat, NRXN1, Dlgap1, Ncam2, and Slc7a10 [15]. In addition, the increase in high GFAP expression in AD is also accompanied by a decrease in the number of astrocytes with low GFAP expression. Therefore, understanding the mechanisms underlying the diminished population of cells exhibiting low GFAP expression in AD is paramount. Inhibiting high GFAP expression may mitigate the abnormal activation of astrocytes observed in AD.

### 2.3. Astrocyte Function

Over the past two decades, research on astrocytes has experienced a remarkable exponential surge, propelled by technological advancements, such as optogenetics, calcium imaging, and the refinement of single-cell sequencing techniques. These innovations have significantly enhanced the comprehensiveness of our understanding of astrocytes. In 2005, Edward S. Boyden and Karl Deisseroth utilized a lentiviral vector to transfect a protein, ChR2 (Channelrhodopsin-2), from the alga Chlamydomonas reinhardtii into neurons, achieving excitatory/inhibitory control of the action potentials and synaptic transmission [42]. Optogenetic technology has been widely applied to the study of neurons and astrocytes since 2010 [43,44]. The optogenetic activation of hippocampal astrocytes increased the intracellular calcium levels of an astrocyte-specific ChR2 transgenic mouse (hGFAP–ChR2 mouse). Furthermore, researchers found that the optogenetic activation of astrocytes could increase the release of ATP from astrocytes, thereby enhancing excitatory synaptic transmission in dentate gyrus (DG) granule cells [44]. This finding has helped to gradually highlight the role of astrocytes as a third-party synapse, regulating the E/I imbalance in neural circuits in AD [45]. Adeno-associated viruses (AAVs) enable rapid genetic manipulation; however, astrocyte specificity may be limited in this regard, with high off-target expression in neurons and sparse expression in endothelial cells. This year, a study reported the development of a box with four copies of the miRNA targeting sequence (4 × 6 T), which explicitly triggers transgenic degradation in neurons and endothelial cells. Used in conjunction with the GfaABC1D promoter, the 4 × 6 T miRNA targeting sequence increased the astrocyte specificity of Cre with a virus reporter gene in mice across various serotypes, from 50% to 99%. When employed with multiple miRNA targeting sequences, it effectively decreased the expression in numerous off-target cell groups for various recombinant enzymes and reporter genes [46].

Despite the growing recognition of astrocytes’ significance in the functioning of the nervous system, their precise roles remain elusive, particularly regarding the mechanisms implicated in neurodegenerative diseases, such as AD. Many fundamental aspects of their biology are still mysterious. In the 1970s, broadly termed “reactive astrocytes”, an increase in the expression of the intermediate filament protein, GFAP (glial fibrillary acidic protein), was discovered [47], which is also a significant characteristic of astrocytes in AD. However, in recent years, astrocyte-specific proteins, such as Clu, have been identified as risk factors in AD. The possible mechanisms of Clu changes in regard to the risk factors caused by astrocyte activation mainly include lipid metabolism [48], glutamate uptake, GABA dysregulation [49], and inflammatory responses [50] in the central nervous system. These changes in astrocytes can influence the pathological features of AD, such as the deposition of extracellular amyloid beta fibril plaques [51,52,53,54], neuroinflammatory responses [19,54,55,56], and neurofibrillary tangles composed of the hyperphosphorylated tau protein [51,52,53]. Understanding the specific risk factor, Clu, in regard to the astrocytes of AD can help us better comprehend the pathogenesis of AD and potentially identify specific targets for detection and treatment.

## 3. The Astrocyte Risk Factor, Clu, in AD

Apolipoprotein J is named Clu, due to its ability to aggregate blood cells in vitro. Given that Clu is specifically expressed in astrocytes, it may serve as a potential risk factor for early-onset AD [15,26,27,28]. Astrocytes mainly express Clu in the nervous system, its function in the nervous system requires further exploration. Under normal physiological conditions, astrocytes synthesize and secrete Clu extracellularly. Two distinct Clu protein isoforms are described in mammalian cells: one with a pro-survival function, a conventional secretory isoform (pre-secretory and mature secretory Clu, referred to as psClu and sClu, respectively), and, another intracellular isoform, found in the cytoplasm (pre-nuclear Clu, pnClu), existing as a mature nuclear clusterin (nClu) of approximately 55 kDa in basal cells and the cell nucleus [57]. Early in 1990, researchers pointed out a significant increase in Clu mRNA and protein levels in the brain regions of AD patients [58]. In clinical cohort studies involving over 16,000 and 224 individuals, AD patients were found to exhibit mutations in the *Clu* gene through the use of whole-genome sequencing [59,60]. Additionally, other experimental studies reported that changes in Clu expression are related to AD [29,30]. Using a large multicenter dataset of 15,239 subjects, a meta-analysis has confirmed Clu as an AD susceptibility locus in European ancestry populations [61]. Clinical research not only demonstrates variations in Clu expression in AD, but also indicates a genetic association between Clu’s variations and AD. Plasma Clu levels are correlated with brain atrophy, baseline disease rates, disease severity, and rapid clinical progression in AD patients, highlighting the importance of Clu in the pathogenesis of AD [62]. PET imaging also reveals a positive correlation between elevated plasma Clu concentrations in AD patients and the typical pathological feature, the Aβ load [62]. In basic research using APPswe/PS1 mouse models, increased Clu expression is observed [63] and the *Clu* gene is significantly upregulated upon exposure to Aβ (Table 1). Thus, alterations in the expression level of the Clu protein, expressed in astrocytes, may serve as a risk factor beyond the GFAP, elucidating the role of Clu in the pathogenesis of AD and contributing to determining whether targeting Clu protein levels is a viable therapeutic option for treating AD. Clusterin is a multifunctional apolipoprotein, and Clu’s involvement in the pathogenesis of AD may encompass lipid transport, Aβ deposition and clearance [52,64], tau pathology [65], and neuroinflammatory responses [27,66], which affect the blood–brain barrier (BBB) in AD. The regulatory mechanism of Clu in regard to the BBB in AD is complex and involves multiple pathways, including: (1) through its interaction with Aβ, Clu is able to bind to Aβ and mediate Aβ clearance by interacting with low-density lipoprotein-related protein 2 (LRP2), myeloid cell trigger receptor 2 (TREM2), or heparan sulfate (HS). This effect helps to reduce the deposition of Aβ in the brain, thereby mitigating damage to the BBB. (2) Affecting microglial function, the binding of Clu to TREM2 can promote the uptake and clearance of Aβ by microglia. The activation of microglia helps maintain the integrity of the BBB and reduces damage to the BBB as a result of the inflammatory response. (3) The regulation of the inflammatory response; the aggregation of Aβ in the brain of AD patients will trigger neuroinflammation, leading to the destruction of the BBB. Clu may inhibit the hydrolysis of the tau protein in lysosomes by binding to the hyperphosphorylated tau protein, resulting in tau aggregation and cell rupture and death. However, this effect may also exacerbate the inflammatory response, further damaging the BBB.

### 3.1. Lipid Transport

AD is closely associated with abnormal lipid metabolism [67,68]. A variety of genetic risk factors for AD are involved in different aspects of lipid metabolism, including lipid trafficking, lipid synthesis, and lipid signaling. Peripheral lipid modulators, including metabolic and vascular risk factors, also alter the risk of AD [67,69,70]. Among the genes at highest risk in terms of sporadic LOAD, APOE, TREM2, Clu, PICALM, ABCA1, and ABCA7 are all directly involved in lipid trafficking or metabolism. Clu is also involved in lipid transport and metabolism in the brain and periphery [70,71].

The brain is a lipid-rich organ, harboring lipids within the cell membranes and myelin sheaths of axons. While all brain cells produce some lipids, the majority of lipids are made in glial cells and transported to neurons. The insolubility of lipids means they cannot be transported between non-adjacent cells unless they are dissolved, which is achieved by enveloping lipids in proteins to form lipid-containing particles within cells, allowing lipids to be transported in soluble lipoprotein particles. Through a series of lipoprotein receptors utilizing receptor-mediated endocytosis (RME), these particles bind to and internalize their cargo into cells, with Clu being a major brain cholesterol transport lipoprotein [57]. As early as the 1990s, Clu was shown to exist in lipid particles secreted by the liver and brain [72]. In plasma, the primary binding partner for Clu is apolipoprotein A1, a major plasma lipoprotein, whose primary source of secretion is the liver. In the brain, the primary source of secreted Clu is astrocytes. Evidence suggests clusterin participates in the clearance of Aβ from the brain by binding to lipoprotein receptors [73]. However, Clu may serve as a potential target protein implicated in the abnormal lipid metabolism observed in AD, which is mediated by Clu [70,71]. Metabolic coordination between neurons and astrocytes is crucial for brain health. However, the coupling of lipid metabolism between neurons and astrocytes, especially in response to neuronal activity, remains largely uncharacterized. The role of Clu, as a significant brain cholesterol transport lipoprotein, in the lipid metabolism abnormalities in AD requires further exploration [74,75,76].

### 3.2. The Pathological Features of AD (Aβ Deposition and Clearance, Tau Pathology)

Clu, encoded by the *Clu* gene, is a multifaceted protein implicated in AD pathogenesis. It interacts with current clinical biomarkers, such as amyloid-β (Aβ), tau, and neuroinflammatory markers. Clu is one of the strongest AD risk factors identified through genome-wide association studies (GWAS), particularly the rs11136000 variant [77,78]. This variant is linked to altered Clu expression, affecting Aβ metabolism and neuroinflammation. Mechanistically, Clu binds to Aβ oligomers, facilitating their clearance via chaperone activity [79]. However, in AD progression, chronic inflammation may drive Clu toward a pro-amyloidogenic role, exacerbating plaque formation. Clu modulates Aβ aggregation and clearance. CSF Aβ42 levels (a hallmark of AD) correlate with Clu overexpression in early-onset AD, suggesting that compensatory mechanisms occur to counteract Aβ toxicity. Plasma Clu levels are elevated in AD and correlate with the Aβ burden, potentially serving as a co-biomarker to improve the specificity of blood tests for amyloid pathology. Clu interacts with tau proteins, influencing their phosphorylation and aggregation. Elevated CSF Clu levels are associated with increased p-tau181 and p-tau217, markers of neurofibrillary tangle formation. Clu and APOE (another major AD risk factor) synergistically regulate lipid metabolism and Aβ clearance. Carriers of both Clu risk alleles and APOE-ε4 exhibit accelerated Aβ deposition and worse cognitive decline, reflected in their biomarker profiles. Although Clu is not yet a stand-alone diagnostic biomarker, its use in combination with Aβ42, p-tau, and NfL can enhance early detection and differential diagnosis.

Aβ is a series of peptides formed by the cleavage of the amyloid precursor protein (APP) by β-secretase (BACE1) and γ-secretase. BACE1 cleaves APP, releasing soluble extracellular domains, sAPPβ and CTFβ. Moreover, γ-secretase cleaves CTFβ, producing different lengths of amyloid beta peptides and the AICD fragment. The appearance of β-amyloid plaques (Aβ) in the brain, extracellular protein deposits, is a major hallmark of AD. These plaques are formed by the spontaneous aggregation of monomeric Aβ into soluble Aβ oligomers, which then aggregate to form insoluble fibrils. Clu expression can alter soluble and insoluble Aβ levels, and the loss of Clu leads to significant changes in the dynamic pools of soluble and insoluble Aβ [64,80,81,82]. Furthermore, Clu is positively correlated with Aβ_42_ and insoluble Aβ40 [83]. Convincing evidence of Clu’s role in Aβ aggregation suggests that the physical interaction between clusterin and Aβ plays a crucial role in the pathogenesis of AD. Clu is a natural partner interacting with Aβ, preventing its aggregation and toxicity [79,84]. In in vitro experiments, Clu has been shown to exhibit significant interaction activity with soluble Aβ (Aβ_1–40_ and Aβ_1–42_). Additionally, the formation of complexes significantly prevents the aggregation of Aβ and soluble Aβ [85]. Clu is capable of forming high-molecular-weight complexes with Aβ and, through its interaction with low-density lipoprotein receptor-related protein 2 (megalin, LRP2) on the cell surface, facilitates the internalization and degradation of Aβ [86,87]. Furthermore, research suggests that the overexpression of Clu can decrease Aβ deposition and improve synaptic transmission in neurons, including reducing the decrease in the mEPSC current in neurons [80]. However, studies using APP transgenic mice with Clu knockout reveal that while the knockout of Clu does not alter the total Aβ content in the mouse brain, it can reduce the formation of Aβ plaques and alleviate the neurofibrillary atrophy caused by Aβ deposition [64,88]. The conditional knockout of BACE1 can reversibly eliminate amyloid plaques formed in adult 5xFAD mice and, similarly, BACE1-deficient astrocytes show a significantly increased uptake of HiLyte Fluor-555 labeled Aβ42 [52]. Clusterin contributes to enhancing the uptake of Aβ in BACE1-knockdown astrocytes, and the loss of BACE1 in 5xFAD mice improves the expression of Clu, while knocking down Clu in BACE1-null astrocytes reduces the uptake of Aβ42 [52]. The lack of BACE1 in astrocytes leads to the increased activation of pP38 and pERK1/2, phosphorylating and activating downstream molecules (such as Clu) mediated by Jun and AP-1, which is related to the enhanced clearance of Aβ by astrocytes. The insulin receptor (IR) is a substrate for BACE1 in the liver, and the IR, composed of heterodimeric dimers, α and β subunits, with the membrane anchored by the β subunit, is identified as a substrate for BACE1. When BACE1 is deleted, the IRβ subunit is no longer cleaved, and the retained IRβ subunit is available to enhance P38, ERK1/2, and MAPK activity [52]. This increased MAPK pathway activates AP-1 transcription to elevate Clu levels and improve the amyloid beta internalization in reactive astrocytes lacking BACE1.

Clu expressed explicitly in astrocytes can act as a ligand for TREM2 expressed in microglia. The overexpression of wild-type TREM2 is sufficient to enhance the uptake of Clu in heterologous cells, while TREM2 disease variants impair this activity. The conditional knockout of Trem2 in microglia shows reduced internalization of Clu. The β-amyloid (Aβ) binds to the lipid–protein Clu, and this complex is efficiently absorbed by microglia in a TREM2-dependent manner. Macrophages from human subjects carrying the AD-associated TREM2 variant show reduced uptake of the Aβ–Clu complex [89]. Additionally, the expression of Clu is also regulated by SORL1, and the loss of SORL1 in the astrocyte leads to increased Aβ production and elevated Clu RNA levels in astrocytes. APP is a product of SORL1, which has a role as a neuronal retrotranslocase, and the loss of SORL1 leads to the retention of APP in endosomes, resulting in increased Aβ production [74].

In addition to Aβ, the second neuropathological hallmark of AD is the accumulation of intracellular tau tangles. The phosphorylation of the tau protein increases before the formation of noticeable tangles. The aggregation and propagation of the tau protein have also garnered increasing attention from researchers. The tau protein itself is a microtubule-binding protein that plays a crucial role in physiological conditions. A key factor prompting the transition of this protein from a physiological to a pathological state is the excessive phosphorylation of the tau protein. The tau protein is prone to phosphorylation and has many phosphorylation sites, primarily serine^+^ proline (SP) and threonine^+^ proline (TP). There are also numerous kinases capable of phosphorylating the tau protein, making the study of the mechanism of tau protein overphosphorylation challenging [90]. In AD, Clu co-localizes with tau aggregates. The absence of Clu exacerbates tau pathology in a mouse model of tauopathy, and Clu significantly inhibits the formation of tau fibrils [65]. But other researchers have also shown that the upregulation of Clu may enhance tau seeding and potentially accelerate the spread of tau pathology. This disparity may be due to contrasting phenotypes of intracellular vs. extracellular tau and the fact that seeding was examined in one context, while fibril formation was quantified in the other. As a molecular chaperone, the tau/Clu complex enters receptor cells through endocytosis, disrupting the endolysosomal compartment and transferring to the cytoplasmic compartment, where they propagate endogenous tau aggregation [91]. Furthermore, Clu can affect the level of tau phosphorylation, exacerbating tau phosphorylation through the autophagic pathway [74]. Microtubule-associated protein 1 light chain 3 (MAP1LC3) is a homolog of Atg8 in mammals. LC3/Atg8 is cleaved at the C-terminal by Atg4 with protease activity, generating the cytoplasmic form, LC3-I. During autophagy formation, cytoplasmic LC3-I participates in a ubiquitin-like reaction, catalyzed by Atg7 and Atg3 (corresponding to ubiquitin-activating enzyme E1 and ubiquitin-conjugating enzyme E2, respectively), resulting in the conjugation of phosphatidylethanolamine (PE) and the formation of the lipidated form, LC3, also known as LC3-II. It attaches to the membrane of the autophagosome, serving as a structural protein in the autophagosome, namely autophagosome membrane-bound LC3-II. During degradation, LC3-II on the outer membrane of the autophagosome is cleaved by cysteine protease Atg4B, producing LC3-I for recycling; inner membrane-associated LC3-II is degraded with the encapsulated contents in the lysosome. This process leads to a low level of LC3 in the autolysosome [92]. Clusterin participates in autophagosome biogenesis through its interaction with ATG8E (MAP1LC3A) as a chaperone protein. Clu co-localizes with LC3 by binding to the LIR motif on the autophagosome membrane, enhancing the stability of the LC3–Atg3 heterocomplex and LC3 lipidation, thereby promoting autophagosome biogenesis and autophagy activation [93]. The autophagy-related pathway is associated with SORL1 loss, and an SORL1 deficiency also increases phosphorylated tau levels. A recent study reported that SORL1 KO neurons exhibit autophagic flux defects, suggesting a potential impact of disrupted autophagy on Clu levels. Treatment with 100 mM of trehalose (a presumed autophagy enhancer) significantly affected the levels of p-tau. After 72 h of treatment, trehalose reduced the elevated p-tau levels without similarly affecting the Aβ levels [74].

## 4. Scientific Hypothesis: The Possible Regulatory Mechanisms of Clu in Regard to Astrocyte Risk Factors in AD

### 4.1. The Neuronal Circuit Excitability/Inhibition Imbalance

As a risk factor for AD, Clu also may be related to the imbalance of excitation and inhibition in neural circuits. One possible mechanism is directly regulating the neurotransmitters, glutamate and GABA, in neurons to modulate the neural circuit excitation and inhibition balance. From a neural network perspective, viewing excitation and inhibition as a single entity, if the level of excitation exceeds that of inhibition, the activity will increase until the circuit maximizes its activity-generating capacity or until such activity begins to be marginally increased, absorbing more inhibition than excitation, resulting in a “balanced” state. Conversely, if inhibition exceeds excitation, the activity will decrease until the circuit is stationary or until marginal decreases lead to a more significant reduction in inhibition than excitation, also resulting in an overall balance. This is the simplest conceptualization of the “excitation–inhibition (EI) balance” [94]. The imbalance of excitation and inhibition is associated with increased glutamate (excitatory) signaling or decreased inhibition associated with GABAergic signaling [95]. Since its discovery in the brain more than 70 years ago, GABA has been recognized as a major inhibitory neurotransmitter. GABA has a role in rapid inhibitory synaptic transmission through exocytosis-mediated presynaptic GABA release and postsynaptic low-affinity GABA_A_ receptor activation. Alterations in GABA signaling affect changes in the overall E/I of the neural network, which in turn affects signaling. At present, relevant studies have shown that Clu can participate in the synthesis of GABA, but the effect on GABA receptors has not been reported [96]. GABA synthesis is the main factor regulating the activity of GABA, and GABA is mainly synthesized through the decarboxylation of glutamate or the degradation of putrescine. The decarboxylation of glutamate is mediated by glutamate decarboxylase 65 (GAD65) [97,98] and GAD67 [99,100], which can be expressed at different locations in the same inhibitory neuron. In Clu mutant mice generated in a previous study, it was found that the synthesis of the neurotransmitter GABA in GABAergic interneurons in an AD model was significantly increased, confirming the dysfunction of GABAergic neurons and the abnormally high GABA neurotransmitter secretion in this model [49]. Changes in GABA signaling can affect the overall E/I balance in neural networks, thereby influencing signal transmission. And the increase in the synthesis of GABA involving Clu may also be the main reason for the decrease in AD excitability (Figure 2).

Another possibility is that astrocytes indirectly affect the balance of excitation and inhibition in neural circuits. Astrocytes are crucial controllers of excitatory and inhibitory synapses [101]. Astrocytes can dynamically regulate neuronal communication through the uptake of neurotransmitters (such as GABA and glutamate) or by releasing glial transmitters (such as D-serine and ATP). Glutamate signaling mediated by astrocytes involves the release of glutamate and changes in the activity of excitatory amino acid transporters in astrocytes to regulate glutamate levels in the synaptic cleft [102]. Astrocytic glutamate transporters, including the glutamate-–aspartate transporter (GLAST), glutamate transporter 1 (GLT-1), and their human homologs, excitatory amino acid transporter 1 (EAAT1) and 2 (EAAT2), are major transporters for taking up synaptic glutamate to maintain optimal extracellular glutamate levels, preventing accumulation in the synaptic cleft and the associated excitotoxicity [102]. Astrocytes regulate glutamate neurotransmission by taking up glutamate neurotransmitters and release glutamate through exocytosis, primarily related to the vesicular glutamate transporter, vGLUT1 [45,103,104,105]. Recently, single-cell sequencing has identified proteins related to synaptic glutamate exocytosis in astrocytes, including the astroglial glutamate transporter vesicular glutamate transporters, Slc17a7, Slc17a6, Snap25, and Syt1 [45]. It was found that they are strongly expressed not only in glutamatergic neurons, but also in a subset of GS/S100β-positive cells belonging to the GFAP lineage (tdTomato), with distinct separated cell nuclei that do not overlap with neurons, which confirmed the existence of a population of astrocytes involved in synaptic glutamate exocytosis [45,104]. In addition, astrocytes may also influence excitatory transmission in neural circuits, by regulating the glutamate receptors of neuronal glutamatergic synapses. NMDA receptors are receptors for the excitatory neurotransmitter glutamate, and astrocytes release D-serine, which assists in the glutamate activation of NMDA receptors. Moreover, adding the D-serine degradation enzyme to brain slices reduces the activity of NMDA receptors [106,107] (Figure 3).

Astrocytes primarily take up the neurotransmitter GABA through the expression of GABA transporters, including GAT-1 and GAT-3. GAT-3 is the most abundant GAT in astrocytes, located in the astrocyte processes adjacent to synapses and cell bodies, while GAT-1 can be identified in distant astrocyte processes and is more plentifully expressed in neurons. The downregulation of GAT-3 expression increases the GABA concentration in the mouse brain, and astrocyte Ca^2+^ signaling regulates tonic inhibition in a GAT-3-dependent manner [108]. The dysregulation of GABA and astrocytic Ca^2+^ signaling is associated with excessive self-modulation, which can be alleviated by blocking GAT-3 [109]. The activation of GAT-3 leads to an increase in the Na^+^ concentration in hippocampal astrocytes, thereby increasing intracellular Ca^2+^. Additionally, GABA uptake by astrocytic GAT-3 stimulates the release of ATP/adenosine, contributing to the downregulation of excitatory synaptic transmission and providing a mechanism for the homeostatic regulation of hippocampal synaptic activity [110]. GABA inhibitory synaptic transmission by GAT-3 is regulated through TRPA1 and MaxiK channels. A TRPA1 channel-mediated reduction in the resting Ca^2+^ concentration in astrocytes reduces the efficacy of GAT-3-mediated GABA transport, thereby increasing extracellular GABA and enhancing the inhibitory synaptic strength of interneurons. Moreover, in human embryonic kidney 293 cells with the SV40 T antigen (HEK293T), GAT-3 was found to co-immunoprecipitate with MaxiK channels. The interaction between MaxiK channels and GAT-3 opens up the possibility of MaxiK channels’ involvement in the steady state and signal transduction of GABA. L-isoleucine, as a substrate for GAT-3, can also regulate the expression of GAT-3. The exogenous administration of L-isoleucine induces increased GAT-3 expression and functional recovery [111]. Rab11a was identified through an RNA-seq assessment as a gene significantly capable of regulating GAT-3 expression, and the downregulation of Rab11a leads to the upregulation of GAT-3 function [109].

The abnormal activation of astrocytes in AD leads to the increased expression of Clu [52]. The specific activation of astrocytes influences neural circuits, and the aberrant activation of astrocytes also affects the homeostasis of neuronal glutamate and GABA. The abnormal regulation of glutamate uptake and GABA in astrocytes contributes to excitotoxicity in AD. Excessive extracellular glutamate levels in AD lead to neural overexcitation, causing an imbalance in the E/I network in the AD brain. Upon activation, astrocytes release neurotoxic NO or glutamate or downregulate the uptake of extracellular neurotransmitters, ultimately leading to neuronal death [112]. In AD, the downregulation of EAAT1 and EAAT2 expression in astrocytes leads to the accumulation of extracellular glutamate, promoting excitotoxicity and consequent neuronal impairment [113]. Similarly, in AD mouse models, the knockdown of EAAT2 in astrocytes exacerbates cognitive decline [114]. However, no specific mechanism has been identified yet indicating the role of VGluT1 in astrocytes and the impact on the GABA transporter, GAT-3, after the abnormal activation of astrocytes, leading to increased Clu expression. This is a key aspect of Clu in regard to its role as a risk factor, involved in regulating the excitatory–inhibitory imbalance in neural circuits.

### 4.2. The Inflammatory Mechanism

Multiple studies indicate that Clu in astrocytes is involved in inflammatory responses through various direct or indirect mechanisms [50]. The NF-κB pathway stands out as the pathway most closely linked to inflammation, with astrocytes being no exception in this regard. Clu is an inhibitor of the NF-κB pathway, and the nuclear translocation of the activated nuclear factor kappa-light-chain-enhancer of activated B cells (NF-κB) heterodimer is a central step in the activation of astrocytes [50]. The precursor secretory form of clusterin (psCLU) possesses NF-κB regulatory activity. The nuclear translocation of NF-κB in astrocytes is triggered by pro-inflammatory stimuli, such as tumor necrosis factor (TNF)-α, interleukin (IL)-1β, IL-17, reactive oxygen species (ROS), phagocytosis of myelin, and Toll-like receptors (TLRs). In AD, the activation of A1-type astrocytes leads to the release of neurotoxic factors, while the signaling of inflammation is primarily attributed to microglia [26]. Most studies focus on the activation of A1-type astrocytes and their association with the inflammatory mechanisms of microglia. Clu activates primary rat microglia cells, leading to morphological changes in microglial cells. Furthermore, Clu-activated microglial cells secrete the neurotoxic substance, TNF-α, in microglia–neuron co-culture models [115], thereby triggering an inflammatory response. Additionally, research indicates that Clu can participate in the C1q-mediated complement pathway [116]. This indirectly demonstrates the association of Clu with the pro-inflammatory function mediated by the A1 astrocyte activation associated with A1-type inflammation. A1-type astrocytes promote the pro-inflammatory function of microglia by secreting IL-6, GM-CSF, and other signaling transduction factors. Microglia secrete IL-1α, TNF-α, and the complement component, C1q (C1q), to induce transcriptional responses in astrocytes, characterized by the production of unidentified neurotoxic factors, reduced phagocytic activity, and the decreased expression of neurotrophic factors [117]. The protective effect of Clu in regard to the activation state of A2 astrocytes against AD remains unclear. However, we speculate that it may be related to the expression of IL-3. Several studies have reported that astrocyte-expressed IL-3 can attenuate the progression of AD and may, therefore, represent a potential target for AD therapy [118,119]. Astrocyte-derived interleukin-3 (IL-3) interacts with microglia to improve the pathology of AD [119]. The deletion of IL3 leads to the decreased expression of Apoe, as well as the inhibition of the expression of genes associated with AD and tissue repair (Spp1, Dkk2, Gpnmb), and microglial immune responses (Clec7a, Igf1, Itgax, Lyz2, Mamdc2, Actr3b, Trem3, Trem1, Ctsg, Ctsw, Cd200r4, Clec4e, Cxcr4, Cxcr6, IL27ra) and genes (Ccl8, Ccl5, Hpse, Lox, Mmp9, Mmp12, Mmp8, Mmp25) that are essential for cell motility, extracellular matrix remodeling, and lysis [118]. Upon recognition of Aβ deposits, microglia increase their IL-3Rα (a specific receptor for IL-3, also known as CD123), and Trem2 signaling increases the responsiveness of microglial IL-3 by upregulating IL-3 receptor α (IL-3Rα). Astrocytes constitutively produce IL-3, which initiates the transcriptional, morphological, and functional programming of microglia, giving them acute immune response programs, enhanced motility, and the ability to aggregate and clear Aβ and tau aggregates [118]. However, there is no research on how IL-3 helps microglia to localize and clear Aβ and tau (Figure 4).

## 5. Discussion

Clu currently ranks as the third major risk factor associated with Alzheimer’s disease (AD), albeit with a considerably smaller effect size compared to the ApoE gene [25,49]. However, recent large-scale genome-wide association studies (GWAS) still indicate that Clu is a risk factor for AD in the broader population. There has been a significant increase in research on the pathogenic mechanisms of Clu in AD recently, but its specific regulatory mechanisms remain unclear. A burgeoning body of data has identified various pathways that may explain the pathogenic mechanisms of Clu expressed in astrocytes in AD. In addition to potentially participating in the aggregation and clearance of Aβ, Clu is also involved in lipid metabolism [48], the imbalance between neural circuit excitation and inhibition [49], brain inflammation [50], and other processes. However, there are still many unresolved issues, including the following aspects: (1) Activated microglia in AD are classified into A1 and A2 states; does Clu, as a risk factor expressed explicitly in AD astrocytes, participate in the activation of A1 and A2 astrocytes in AD? (2) In regard to the mechanism of Clu involvement in the imbalance between neural circuit excitation and inhibition, does Clu regulate astrocytes in regard to the uptake and transport of glutamate and GABA neurotransmitters, and what are the main molecular pathways involved? (3) Both ApoE and Clu regulate abnormal lipid metabolism in AD; what are the differences in their regulatory roles? (4) Can Clu participate in AD inflammatory signaling, primarily related to microglial cells? Additionally, since Clu can stimulate microglial cells, how can neurons be preserved without causing harmful effects to microglial cells? These are essential questions that require further investigation. Ultimately, targeting or customizing specific types of Clu may be necessary to achieve specific therapeutic goals. Understanding whether Clu plays a mechanistic role in the progression of AD is an essential question for the future, and further exploration of its pathological and physiological role in AD will provide a foundation for the targeted diagnosis and treatment of AD, based on the astrocyte-specific expression of Clu.

**Table 1 ijms-26-07271-t001:** Astrocytes and Clu are associated with clinical AD.

Subtypes	Numbers	Features	Reference
Clinical	818	GFAP upregulation at preclinical stage	[7]
Clinical	96	In cognitively normal older adults at risk of AD, plasma GFAP levels are elevated	[4]
Clinical	90	The GFAP increased in the MCI stage of mild cognitive impairment	[8]
Clinical	16,000	CLU mutations in AD	[59]
Clinical	2338	CLU as a risk factor	[120]
Clinical	224	CLU as a risk factor	[60]
Clinical	15,239	Meta-analysis, CLU as a risk factor	[61]
Clinical	749	CLU is associated with the pathological features of AD	[62]
Mice	APPswe/PS1	CLU upregulation	[63]
Mice	Thy-TAU22	CLU upregulation	[63]

## Figures and Tables

**Figure 1 ijms-26-07271-f001:**
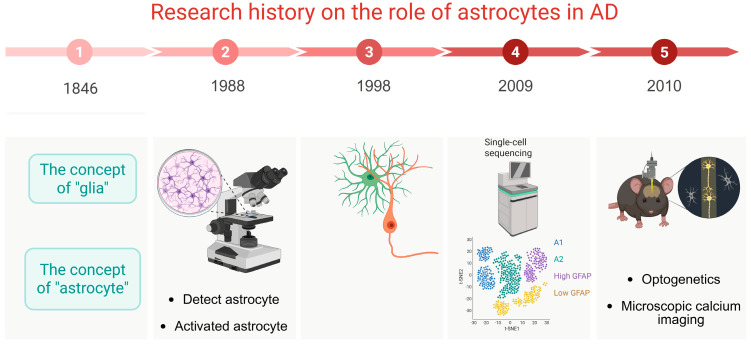
History of the research on astrocytes. In 1846, the concept of glial cells was first proposed. In 1988, the morphology of astrocytes was successfully detected. In 1998, the interaction between astrocytes and neurons was explored. In 2009, single-cell sequencing technology was used to study the different expression clusters of astrocytes in AD. And, in 2010, researchers began to gradually understand mechanisms of astrocytes in the pathogenesis of AD using new technologies, such as optogenetics and calcium imaging.

**Figure 2 ijms-26-07271-f002:**
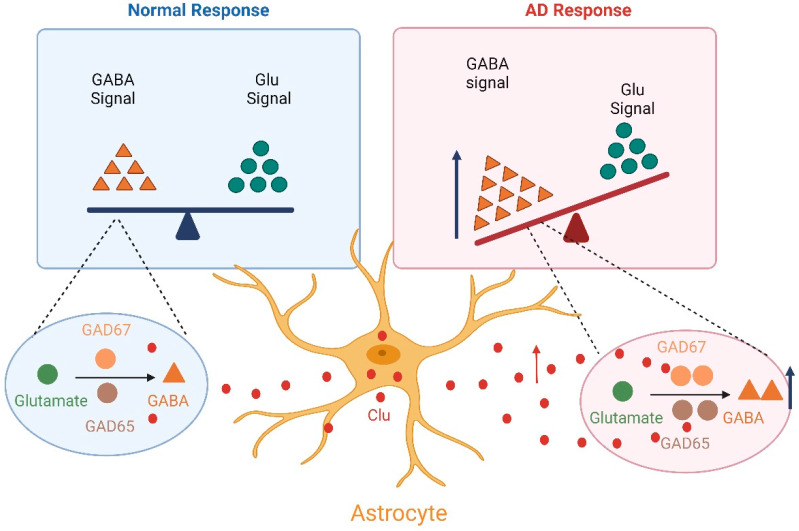
Clu is involved in the imbalance of excitability/inhibition. Aberrant activation of astroglial cells in AD leads to increased expression of Clu, resulting in a significant increase in GAD65 and GAD67, the key enzymes in GABA synthesis. Finally, the excitatory/inhibitory balance of neurons is disrupted.

**Figure 3 ijms-26-07271-f003:**
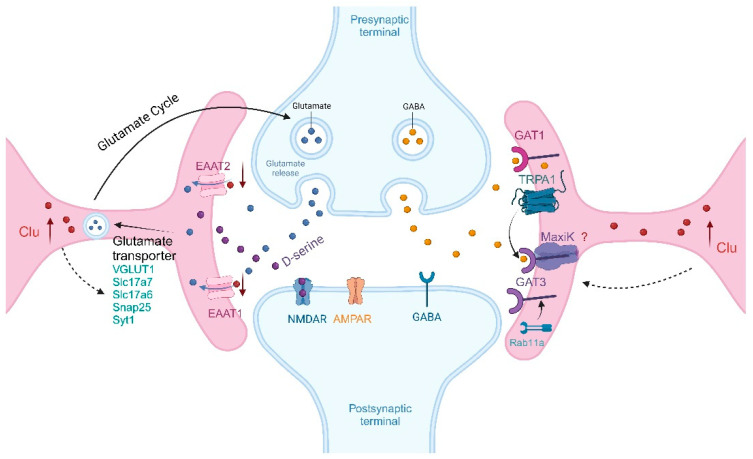
Clu, through astrocytes, indirectly affecting the balance of excitation and inhibition in neural circuits. The increase in the expression of Clu in astrocytes in AD may regulate the excitatory and inhibitory balance of neural circuits through the synthesis and uptake of glutamatergic (left) and GABAergic (right) signaling molecules.

**Figure 4 ijms-26-07271-f004:**
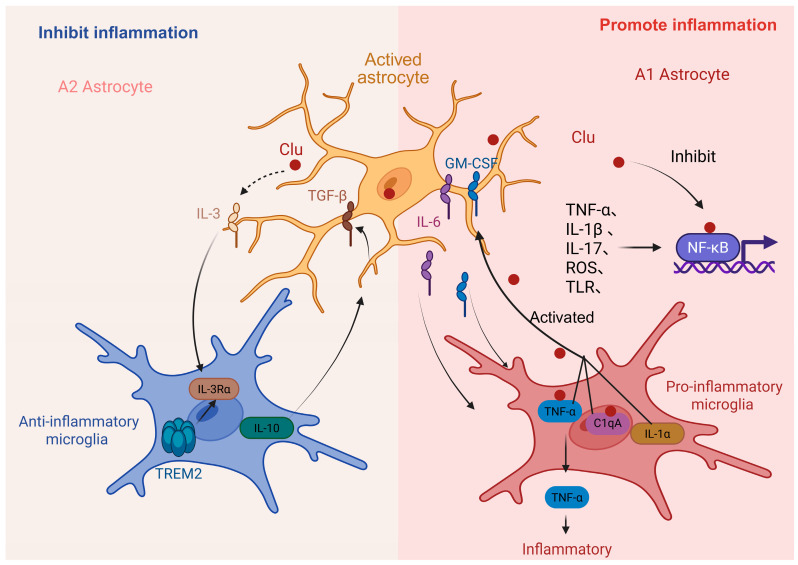
Clu in astrocytes is involved in inflammation in AD. Clu promotes inflammation through the interaction of A1 astrocytes with microglia. In addition, Clu inhibits inflammation through the interaction of A2 astrocytes with microglia.

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
