# Peer review of "Clusterin Regulates the Mechanisms of Neuroinflammation and Neuronal Circuit Impairment in Alzheimer’s Disease"

_ijms, 2025, doi:10.3390/ijms26157271_

Round 1
Reviewer 1 Report
Comments and Suggestions for Authors
Authors prepared a review on clunsterin/ApoJ (Clu) and its imvolvements in neurodegenerative diseases, especially in Alzheimer's disease.
1st of all, authors failed to cite important reports, even though much of the materials in the text coincided with the references as mentioned.
Authors must be specific on the exact pathways of the influences of Clu, since it is like a tipping point for many of the pathogenicity in developing neurodegenerative diseases through not only inflammation.
Figure 3 and 4 are well drawn, but authors must present more detailed pathways, including from the persepective of plasma and BBB.
Authors shoudl also discuss how Clu is related with current clinical biomarkers in neurodegenerative diseases, especially in Alzheimer's disease.
Author Response
Dear reviewer:
We highly appreciate for these valuable and suggestive comments from reviewers. We have tried our best to revise the manuscript entitled “A potential biomarker of preclinical Alzheimer’s Disease: the olfactory dysfunction and its pathogenesis-based neural circuitry impairments” (reference number: ijms-3454356). Those comments are very helpful for revising and improving our manuscript and they also provide important guidance for our research. Regarding the comments, we have studied these comments very carefully, have responded to the reviewers’ questions one by one. Please check all the corrections in the revised manuscript and the responses to the comments are as follows (the corrections and replies are highlighted in red).
The detailed responses to referees’ comments are listed below.
Authors prepared a review on clunsterin/ApoJ (Clu) and its imvolvements in neurodegenerative diseases, especially in Alzheimer's disease.
1st of all, authors failed to cite important reports, even though much of the materials in the text coincided with the references as mentioned.
Thanks very much for this suggestion. We have added cited references.
Authors must be specific on the exact pathways of the influences of Clu, since it is like a tipping point for many of the pathogenicity in developing neurodegenerative diseases through not only inflammation.
Thanks very much for this suggestion. We added the mechanism by which Clu affects lipid metabolism, Aβ, and so on by influencing the blood-brain barrier. Please refer to the line 219-234, line 263-281 in the revised manuscript.
Figure 3 and 4 are well drawn, but authors must present more detailed pathways, including from the persepective of plasma and BBB.
Thanks very much for this suggestion. This has been revised.Please refer to the line 219-234 in the revised manuscript.
Authors shoudl also discuss how Clu is related with current clinical biomarkers in neurodegenerative diseases, especially in Alzheimer's disease.
Thanks very much for this suggestion. We added the relationship of the Clu and current clinical biomarkers in the AD. Please refer to the line 263-281 in the revised manuscript.

Reviewer 2 Report
Comments and Suggestions for Authors
The manuscript entitled „Clusterin regulates the mechanisms of neuroinflammation and neuronal circuit impairment in Alzheimer's disease” by Yihang Yu and coworkers is of interest to the Journal’s Readers but is a little bit poor written. At first sight, we believe that the whole structure of the manuscript should be rethought. As a suggestion, after their large documentation, the authors should think what they want to communicate. Is there a link between Clusterin and AD? If this may exists, then a draft/skeleton of manuscript should show the main ideas. After that they may bring all arguments to support their position.
We mention some unclear parts of the manuscript below.
In Abstract: „As the third-party synapses of the brain network, astrocytes play an important role in maintaining the normal function of neural networks , which contribute to the abnormal function of AD.” Because AD means Alzheimer's disease, „the abnormal function of AD” is strange. Please rephrase.
Then, „. In recent years, numerous studies have shown that Clusterin, as a protein expressed by astrocytes, can participate in the progression of AD.” I seem that „as” is not necessary.
In addition, „The pathogenesis of AD regulated by Clusterin is associated with lipid metabolism, AD pathological features, imbalance in neural circuit excitatory inhibition, and neuroinflammation.” It seems to me that Clusterin generates all pathological features. Is that correct. In other words, if no Clusterin exists, then no AD occurs.
The authors refer to „glial fibrillary acidic protein (GFAP), which is considered a biomarker for reactive astrocyte proliferation.” But what relevance has GFAP? Why they insisted on it?
Page 2/72: „However, the regulatory mechanisms of Clu during the development of AD remain elusive.” Is it about Clu as Clusterin or Clu gene? It is not clear.
Page 4/190: „Apolipoprotein J named Clu due to its ability to aggregate blood cells in vitro.” Clu was introduced not from the beginning but on page 4. In addition, Clu is a protein or a gene? It is not clear.
Page 4/192:„Astrocytes mainly express Clu in the nervous system, while extensive research indicates a cell-protective role of Clu in the heart and kidneys, its function in the nervous system requires further exploration.” If Clu has a protective role in the heart and kidneys, then how can Clu be associated with AD?
Page 5/203: „other experimental studies suggest that changes in Clu expression also increase the risk of AD 29,30.” Are clear evidence for this sentence or only suggestions? Could be a simple association between events, and not a cause-effect relationship.
All these and all information found in this manuscript seems to be only suppositions. The authors should demonstrate the mechanism by which Clusterin regulates the mechanisms of neuroinflammation, as they have written in the title.
Minor points
Page 2, row 46: „Considering that the concentration of GFAP in the blood is related to the degree of amyloid-β (Aβ) and phosphorylation of Tau.” Please , se the typographic mistakes (i.e. amyloid-β (Aβ) and).
Idem row 48: „Aβ and”, etc.
Page 2/65: „However, Clu may also be a new risk factor for AD 15,26-28 .” The authors introduced the Clu gene too abruptly. What is Clu gene?
Page 7/338: 4. Scientific hypothesis: astrocyte risk factors Possible regulatory mechanisms of CLU in AD.
Because of many drawbacks, this manuscript cannot be publicated in the present form and it deserves a severe revision.
Comments on the Quality of English LanguageThere are minor point on English and typography.
Author Response
Dear reviewer:
We highly appreciate for these valuable and suggestive comments from reviewers. We have tried our best to revise the manuscript entitled “A potential biomarker of preclinical Alzheimer’s Disease: the olfactory dysfunction and its pathogenesis-based neural circuitry impairments” (reference number: ijms-3454356). Those comments are very helpful for revising and improving our manuscript and they also provide important guidance for our research. Regarding the comments, we have studied these comments very carefully, have responded to the reviewers’ questions one by one. Please check all the corrections in the revised manuscript and the responses to the comments are as follows (the corrections and replies are highlighted in red).
The detailed responses to referees’ comments are listed below.
The manuscript entitled „Clusterin regulates the mechanisms of neuroinflammation and neuronal circuit impairment in Alzheimer's disease” by Yihang Yu and coworkers is of interest to the Journal’s Readers but is a little bit poor written. At first sight, we believe that the whole structure of the manuscript should be rethought. As a suggestion, after their large documentation, the authors should think what they want to communicate. Is there a link between Clusterin and AD? If this may exists, then a draft/skeleton of manuscript should show the main ideas. After that they may bring all arguments to support their position.
We mention some unclear parts of the manuscript below.
In Abstract: „As the third-party synapses of the brain network, astrocytes play an important role in maintaining the normal function of neural networks , which contribute to the abnormal function of AD.” Because AD means Alzheimer's disease, „the abnormal function of AD” is strange. Please rephrase.
Thanks very much for this suggestion. We have revised this following your suggestion. Please refer to the line 11-12 in the revised manuscript.
Then, „. In recent years, numerous studies have shown that Clusterin, as a protein expressed by astrocytes, can participate in the progression of AD.” I seem that „as” is not necessary.
Thanks very much for the comment. This has been revised. Please refer to the line 12 in the revised manuscript.
In addition, „The pathogenesis of AD regulated by Clusterin is associated with lipid metabolism, AD pathological features, imbalance in neural circuit excitatory inhibition, and neuroinflammation.” It seems to me that Clusterin generates all pathological features. Is that correct. In other words, if no Clusterin exists, then no AD occurs.
Thanks very much for the comment. This has been revised. Please refer to the line 13-15 in the revised manuscript.
The authors refer to „glial fibrillary acidic protein (GFAP), which is considered a biomarker for reactive astrocyte proliferation.” But what relevance has GFAP? Why they insisted on it?
Thanks very much for the comment. We've added a description of GFAP. Please refer to the line 39-43 in the revised manuscript.
Page 2/72: „However, the regulatory mechanisms of Clu during the development of AD remain elusive.” Is it about Clu as Clusterin or Clu gene? It is not clear.
Thanks very much for the comment. This has been revised. Please refer to the line 67-69 in the revised manuscript.
Page 4/190: „Apolipoprotein J named Clu due to its ability to aggregate blood cells in vitro.” Clu was introduced not from the beginning but on page 4. In addition, Clu is a protein or a gene? It is not clear.
Thanks very much for the suggestion. We've adjusted the structure of the article so that the introduction to Clusterin is placed at the beginning of the article. In addition, the Clu often refers to clusterin, a name that is applied to both genes and the proteins they code.
Page 4/192:„Astrocytes mainly express Clu in the nervous system, while extensive research indicates a cell-protective role of Clu in the heart and kidneys, its function in the nervous system requires further exploration.” If Clu has a protective role in the heart and kidneys, then how can Clu be associated with AD?
Thanks to your suggestion, the mechanism of heart-derived clusterin versus glial cell-derived clusterin in improving the pathological features of AD is not yet well understood. There may be a multi-organ synergy in the function of Clu, which may also be a new strategy to explore the pathogenesis of AD in the future.
Page 5/203: „other experimental studies suggest that changes in Clu expression also increase the risk of AD 29,30.” Are clear evidence for this sentence or only suggestions? Could be a simple association between events, and not a cause-effect relationship.
Thanks very much for the comment. This has been revised. Please refer to the line 205 in the revised manuscript.
All these and all information found in this manuscript seems to be only suppositions. The authors should demonstrate the mechanism by which Clusterin regulates the mechanisms of neuroinflammation, as they have written in the title.
Minor points
Page 2, row 46: „Considering that the concentration of GFAP in the blood is related to the degree of amyloid-β (Aβ) and phosphorylation of Tau.” Please , se the typographic mistakes (i.e. amyloid-β (Aβ) and).
Thanks very much for the comment. This has been revised. Please refer to the line 48 in the revised manuscript.
Idem row 48: „Aβ and”, etc.
Page 2/65: „However, Clu may also be a new risk factor for AD 15,26-28 .” The authors introduced the Clu gene too abruptly. What is Clu gene?
Thanks very much for the comment. This has been revised. Please refer to the line 67-69 in the revised manuscript.
Page 7/338: 4. Scientific hypothesis: astrocyte risk factors Possible regulatory mechanisms of CLU in AD.
Thanks very much for the comment. I'm sorry, I didn't fully understand what you meant, whether it was suggesting that we change the topic or if there was a grammatical problem with the topic.
Because of many drawbacks, this manuscript cannot be publicated in the present form and it deserves a severe revision.

Round 2
Reviewer 1 Report
Comments and Suggestions for Authors
Authors revised the manuscript as commented previously.
It will be nice to make a table of comparing the changes of all biomarkers with BDNF.
All important references are not updated.
Author Response
Dear reviewer:
Thank you for your insightful comments on our manuscript. We have tried our best to revise the manuscript entitled “A potential biomarker of preclinical Alzheimer’s Disease: the olfactory dysfunction and its pathogenesis-based neural circuitry impairments” (reference number: ijms-3454356).The corrections and replies are highlighted in red.
The detailed responses to referees’ comments are listed below.
It will be nice to make a table of comparing the changes of all biomarkers with BDNF.
Thanks very much for this suggestion. In response to your suggestion, we have now added a new table (Table 2) comparing the changes in key AD biomarkers. (e.g., Aβ, tau, GFAP,ApoE,Clu) alongside BDNF levels. This table provides a clearer overview of their potential interactions and clinical relevance, as you recommended.
All important references are not updated.
Thanks very much for this suggestion. We have added references 121-130.
Reviewer 2 Report
Comments and Suggestions for Authors
The manuscript was severely revised and it has enough merits to be publisjed in the Journal. The manuscript is now well-written and it contains scientific information of interest to the Journal's Readers.
Author Response
Dear reviewer:
Thank you very much for your thoughtful and constructive comments on our manuscript, " A potential biomarker of preclinical Alzheimer’s Disease: the olfactory dysfunction and its pathogenesis-based neural circuitry impairments " (Manuscript ID: ijms-3454356). We sincerely appreciate the time and effort you have dedicated to reviewing our work and providing us with valuable feedback.
Thank you for your support and for helping us enhance our manuscript.